# Triptolide Attenuates Muscular Inflammation and Oxidative Stress in a Delayed-Onset Muscle Soreness Animal Model

**DOI:** 10.3390/ijerph192416685

**Published:** 2022-12-12

**Authors:** Che-Chia Hsu, Chin-Chuan Tsai, Po-Yen Ko, Ting-Hsien Kwan, Ming-Yie Liu, Po-Ting Wu, I-Ming Jou

**Affiliations:** 1Department of Orthopaedics, National Cheng Kung University Hospital, College of Medicine, National Cheng Kung University, Tainan 70428, Taiwan; 2Department of Traditional Chinese Medicine, E-Da Dachang Hospital, Kaohsiung 82445, Taiwan; 3School of Chinese Medicine for Post Baccalaureate, I-Shou University, Kaohsiung 82445, Taiwan; 4Department of Orthopaedics, Ditmanson Medical Foundation Chia-Yi Christian Hospital, Chiayi 60002, Taiwan; 5Department of Environmental and Occupational Health, College of Medicine, National Cheng Kung University, Tainan 70428, Taiwan; 6Medical Device Innovation Center, National Cheng Kung University, Tainan 70428, Taiwan; 7Department of Orthopaedics, College of Medicine, National Cheng Kung University, Tainan 70428, Taiwan; 8Department of Biomedical Engineering, National Cheng Kung University, Tainan 70428, Taiwan; 9Department of Biochemistry and Molecular Biology, College of Medicine, National Cheng Kung University, Tainan 70428, Taiwan; 10Department of Orthopaedics, E-Da Hospital, Kaohsiung 82445, Taiwan; 11School of Medicine, College of Medicine, I-Shou University, Kaohsiung 82445, Taiwan; 12GEG Orthopedic Clinic, Tainan 74543, Taiwan

**Keywords:** delayed-onset muscle soreness, triptolide, oxidative stress, inflammation, nitric oxide, myeloperoxidase

## Abstract

Delayed-onset muscle soreness (DOMS) is associated with exercise-induced muscle damage and inflammation, which is mainly caused by prolonged eccentric exercise in humans. Triptolide, an extract from the Chinese herb Tripterygium wilfordii Hook F, has been used for treating autoimmune and inflammatory diseases in clinical practice. However, whether triptolide attenuates acute muscle damage is still unclear. Here, we examined the effect of triptolide on carrageenan-induced DOMS in rats. Rats were injected with 3% of carrageenan into their muscles to induce acute left gastrocnemius muscular damage, and triptolide treatment attenuated carrageenan-induced acute muscular damage without affecting hepatic function. Triptolide can significantly decrease lipid hydroperoxide and nitric oxide (NO) levels, proinflammatory cytokine production, and the activation of nuclear factor (NF)-ĸB, as well as increase a reduced form of glutathione levels in carrageenan-treated rat muscles. At the enzyme levels, triptolide reduced the inducible nitric oxide synthase (iNOS) expression and muscular myeloperoxidase (MPO) activity in carrageenan-treated DOMS rats. In conclusion, we show that triptolide can attenuate muscular damage by inhibiting muscular oxidative stress and inflammation in a carrageenan-induced rat DOMS model.

## 1. Introduction

Delayed-onset muscle soreness (DOMS) is a symptom of prolonged eccentric exercise-associated muscle injury and inflammation [1]. In DOMS, muscle fiber microtrauma leads to an adaption of muscle to prevent damage, thereby causing soreness [1]. DOMS is commonly observed in athletes and people who are undergoing eccentric exercise [1]. Emerging evidence indicates that oxidative stress plays a central role in the initiation and perpetuation of inflammatory responses, which are involved in the pathophysiological changes of various diseases, including cardiovascular disease, diabetes, and cancers [2]. Over-production of free radicals (superoxide anions, hydroxyl radicals, and nitric oxide) and reduced levels of circulating antioxidants (reduced form of glutathione, GSH) predominantly account for muscular oxidative stress [3]. In order to develop effective therapeutics for DOMS, oxidative stress and inflammation in DOMS should be considered.

At the molecular level, several oxidative stress-associated proteins may be involved in this distinct process. Inducible NO synthase (iNOS)-produced nitric oxide (NO) is one of the reactive nitrogen species that has been reported to be involved in the pathogenesis of both oxidative stress and inflammation [4,5]. Furthermore, NO enhances the activity of nuclear factor (NF)-ĸB [6,7] and the generation of pro-inflammatory cytokines, including tumor necrosis factor-α (TNF)-α and interleukin (IL)-1β [8,9]. Myeloperoxidase (MPO), a highly expressed protein in neutrophils, plays a primary role in leukocyte-mediated host defenses [10]. MPO is known to contribute to oxidative damages in many inflammatory disorders [11,12] and can increase iNOS activity by blocking NO-exerted feedback action on iNOS [13]. Up to the present, these molecules have not been studied together in DOMS before.

Tripterygium wilfordii Hook F (TWHF), a traditional Chinese herb, has a long history of being used to treat a variety of inflammatory disorders that dates back many centuries [14,15,16]. Triptolide, as well as extract from TWHF, have been tested in various models of inflammatory and autoimmune disorders, including nephritis, asthma, arthritis, and neurodegenerative disorders, and they have been found to modulate a variety of inflammatory mediators [17]. For instance, triptolide can exert its anti-hepatofibrotic effects in animal models of liver fibrosis and can inhibit the NF-κB signaling pathway in hepatic stellate cells [18]. Wang et al. also demonstrated the anti-inflammatory effects of triptolide through inhibiting the NF-κB signaling pathway in an LPS-induced acute lung injury murine model [19]. Triptolide can inhibit superoxide anion and reactive oxygen species’ (ROS) production in murine peritoneal macrophages by down-regulating NF-κB activation [20,21] and suppress ROS generation and p38 mitogen-activated protein kinase (MAPK) activation in C5b-9-induced podocytes [22]. However, the application of triptolide in acute muscle damage remains elusive. The aim of the present study is to examine the therapeutic effect of triptolide on a rat model of acute muscle damage and examine its possible mechanism.

## 2. Materials and Methods

### 2.1. DOMS Animal Model

Male *Sprague Dawley* rats were obtained from and housed in our institution’s laboratory animal center. They were housed individually in a room with a 12/12-h light/dark cycle and with central air conditioning (25 °C, 70% humidity). The animal care and experimental protocols were in accordance with institutional guidelines, which were approved by the National Cheng Kung University Animal Center (Approval number: 105035). For inducing acute muscular damage in rats, carrageenan, purchased from Sigma (Cat.#C1013, St Louis, MO, USA), was injected in the amount of 100 μL of 3% carrageenan into rats’ left gastrocnemius muscle under isoflurane (3%) anesthesia, as published previously [23].

### 2.2. In Vivo Study Design

Thirty rats were divided into five groups. In the normal (N) group, rats received 100 μL of saline by gastrocnemius muscle injection into the left limbs and 500 μL of saline by intraperitoneal (i.p.) injection; in the control (C) group, rats received 100 μL of 3% carrageenan (Sigma-Aldrich, St Louis, MO, USA) by gastrocnemius muscle injection into the left limbs and 500 μL of saline by i.p. injection; and in the CT30, 100, and 300 groups, rats received 100 μL of 3% carrageenan by gastrocnemius muscle injection into the left limbs and 500 μL of triptolide (Cat.#T3652, Sigma-Aldrich, St Louis, MO, 30, 100, and 300 mg/kg) by i.p. injection. Six hours after the above treatments, a pan behavior test was performed in each rat. After this, rats were euthanized, and their serum and muscle tissue samples were collected for further analyses.

### 2.3. Pain Behavior Test

Rat muscle pain was assessed using an incapacitance meter (IITC, Inc., Woodland Hills, CA, USA), which is a behavioral analysis assay that measures weight bearing on the hindlimbs while the animal is in an induced rearing posture. In brief, it is an in-capacitance meter that consists of two scales and a specialized cage to encourage a rearing posture. Hindlimb weight bearing was tracked and recorded over a 3 min period. These data were transformed into weight distribution by dividing the weight on the right limb by the total weight of both hindlimbs [24].

### 2.4. Serum Biochemical Analysis

Serum biochemical changes, including lactate dehydrogenase (LDH), creatine phosphokinase (CPK), glutamate oxaloacetate transaminase (GOT), and glutamate-pyruvate transaminase (GPT), were assessed by a serum biochemical analyzer (DRI-CHEM 3500s; Fujifilm, Kanagawa, Japan).

### 2.5. Western Blot

We used a nuclear extraction kit (Cat.#NXTRACT, Sigma, Inc., St. Louis, MO) to separate nuclear and cytosolic proteins in the muscle tissues. Fifty micrograms of proteins were used to perform sodium dodecyl sulfate polyacrylamide gel electrophoresis, and the blot was obtained in transferred nitrocellulose sheets (Cat.#NBA085B001EA, NEN Life Science Products, Inc., Boston, MA, USA). After blocking in 5% non-fat skim milk, NF-ĸB p65 (Cat.#SC-8008, Santa Cruz Biotechnology, Dallas, TX, USA), phospho-IĸB (Serine 32, Cat.#SC -8404, Santa Cruz Biotechnology, Dallas, TX), iNOS (Cat.#ab283655, Abcam, Cambridge, UK), and GAPDH (Cat.#32233, Santa Cruz Biotechnology, Dallas, TX, USA) antibodies (dilution 1:1500) were added and incubated at 4 °C overnight. Then, secondary antibodies conjugated with alkaline phosphatase (dilution 1:2000) (Cat.#111-055-003 and 111-055-146, Jackson ImmunoResearch Laboratories, Inc., Philadelphia, PA, USA) were added onto the blots. Immunoblots were observed after bromochloroindolyl phosphate/nitroblue tetrazolium solution was added (Cat.#5410-0013, Kirkegaard and Perry Laboratories, Inc., Baltimore, MD, USA) [25].

### 2.6. Measuring Lipid Peroxidation Level in Muscle

Muscle tissues were homogenized in Milli-Q water. Tissue homogenates (500 μL) were centrifuged at 2500× *g* for 10 min at 4 °C. Five hundred μL of supernatant was sent for lipid hydroperoxide measurement using a kit (Cat.#ab133085, Lipid Hydroperoxide Assay Kit, Abcam, Cambridge, UK), according to the manufacturer’s instruction.

### 2.7. Measuring Muscular Glutathione (GSH) Level

Muscle tissues were homogenized in 10% of ice-cold trichloroacetic acid and then centrifuged at 3000 rpm for 10 min. Supernatant (500 μL) was added to 2 mL of 0.3 M Na_2_HPO_4_ solution. After adding 200 μL of dithiobisnitrobenzoate solution, the absorbance at 412 nm was measured immediately, as described previously [26].

### 2.8. Measuring Pro-Inflammatory Cytokines in Muscles

Levels of TNF-α and IL-1β in muscle tissue homogenates were measured using enzyme-linked immunosorbent assay (ELISA) kits (Cat.#DY510 and DY501, DuoSet; R&D Systems Inc., Minneapolis, MN, USA), according to the manufacturer’s instruction.

### 2.9. Measuring Nitrite Concentration

The amounts of nitrite in muscle tissue were measured by incubating 100 μL of tissue homogenates with 100 μL of Griess solution and using a spectrophotometer at an absorbance of 550 nm. Further, sodium nitrite was prepared as the standard solution, as described previously [27].

### 2.10. Histological Evaluation of Muscular Injury

Muscle tissues were fixed in 4% formaldehyde. Tissue fragments were washed in phosphate buffer, dehydrated in graded concentrations of ethanol, and then embedded in paraffin. From each tissue, 4 µm sections were prepared and stained with hematoxylin and eosin to evaluate muscular morphology. The inflammation score was evaluated as published previously, where 0: none; 1: giant cells, lymphocytes, plasma cells; 2: giant cells, eosinophil, neutrophil; 3: many inflammatory cells [28].

### 2.11. Measuring Hepatic MPO Activity

Muscle tissues were homogenized in 20 mM phosphate buffer (pH 7.4) and then centrifuged (13,000 rpm for 10 min at 4 °C). The pellet was resuspended in 1 mL of 50 mM phosphate buffer containing 0.5% hexadecyltrimethylammonium bromide. The suspension was subjected to four cycles of freezing and thawing, and then centrifuged (13,000 rpm for 5 min at 4 °C). Supernatant (0.5 mL) was mixed with tetramethylbenzidine (0.5 mL) and incubated for 1 min. The reaction was stopped by adding 0.5 mL of 2 N H_2_SO_4_. The spectrophotometer was then used for measurement at an absorbance of 405 nm. MPO activity is shown as the absorbance at 405 nm/min/mg protein, as published previously [29].

### 2.12. Statistical Analysis

Data are means ± standard deviation (SD). Group comparisons were performed using SPSS statistical software (SPSS Institute, Chicago, IL, USA). One-way ANOVA followed by Dunnett’s multiple comparison tests was used for among-group comparison. Statistical significance was set at a *p* value less than 0.05.

## 3. Results

### 3.1. Triptolide Reduces Pain, Serum LDH and CPK Levels without Affecting Hepatic Function in the DOMS Rat Model

To examine the effects of triptolide on muscle damage and hepatic function, serum LDH, CPK, GOT, and GPT levels were determined in carrageenan-treated rats. Carrageenan increased serum LDH (Figure 1A) and CPK (Figure 1B) levels, whereas triptolide significantly decreased their levels at a dose of 100 or 300 mg/kg compared with the carrageenan control group. However, neither carrageenan nor triptolide altered the levels of GOT (Figure 1C) and GPT (Figure 1D). In addition, carrageenan decreased the percentage of weight bearing in rats, whereas triptolide significantly increased this at a dose of 300 mg/kg compared with the C group (Figure 2).

### 3.2. Triptolide Suppresses NO, iNOS, Reactive Nitrogen Species (RNS), and MPO Expression and Oxidative Stress in the DOMS Rat Model

To examine the involvement of reactive nitrogen species (RNS) in triptolide-exerted muscular protection, NO levels were determined in carrageenan-treated rats. Carrageenan significantly increased muscular NO concentration, whereas triptolide at a dose of 300 mg/kg decreased it compared with the carrageenan control group (Figure 3A). In addition, carrageenan increased the expression of iNOS, and triptolide at a dose of 300 mg/kg significantly decreased this when compared with the carrageenan control groups (Figure 3B). To assess the role of MPO in triptolide-associated anti-oxidative effects against carrageenan, muscular MPO activity was determined. Carrageenan significantly increased the activity of muscular MPO compared with the normal group. Triptolide at doses of 100 and 300 mg/kg significantly reduced the MPO activities compared with the carrageenan control group (Figure 3C).

To examine the involvement of oxidative stress in triptolide-treated DOMS rats, muscular lipid hydroperoxide and GSH levels were determined. Carrageenan increased lipid hydroperoxide but decreased GSH levels in carrageenan-treated muscles compared with normal controls; however, triptolide significantly reversed these patterns (Figure 3D,E).

### 3.3. Triptolide Decreases Expression Levels of TNF-α, IL-1β, Nuclear NF-ĸB, and Leukocyte Infiltration in the DOMS Rat Model

To examine the effect of triptolide on carrageenan-induced muscular inflammation, muscular pro-inflammatory cytokine levels and the expressions of NF-ĸB and IĸB were determined. Carrageenan significantly increased muscular TNF-α (Figure 4A) and IL-1β (Figure 4B) levels compared with the normal group; however, triptolide significantly attenuated carrageenan-induced pro-inflammatory cytokine expressions at doses of 100 and 300 mg/kg compared with those in the carrageenan-treated group (Figure 4A,B). Furthermore, both nuclear NF-ĸB (Figure 4C) and p-IĸB (Figure 4D) expressions were increased compared with the normal group, whereas triptolide significantly reduced these expressions at doses of 100 and 300 mg/kg compared with the carrageenan control groups (Figure 4C,D). In the histological examination, increased inflammatory cell infiltration into muscles was observed in the carrageenan control group (Figure 5B); however, mild infiltration was found in muscle tissues from rats treated with 30 (Figure 5C) and 100 (Figure 5D) mg/kg of triptolide. Very little inflammatory cell filtration was found in the muscle tissues from the normal group (Figure 5A) and the group treated with 300 mg/kg of triptolide (Figure 5E). The groups treated with 100 and 300 mg/kg of triptolide had significantly lower inflammation scores than the carrageenan control group (Figure 5F).

## 4. Discussion

In this study, we demonstrate that the natural product extract triptolide attenuates pain levels, as well as muscular inflammation and oxidative stress, without affecting hepatic function in a DOMS rat model. Triptolide decreased muscular proinflammatory cytokine release and neutrophil infiltration. Triptolide can also reduce muscular oxidative stress, NO production, and iNOS expression in carrageenan-treated rat muscles. Furthermore, triptolide inhibits the activation of NF-ĸB and muscular MPO activity. We suggested that triptolide could protect carrageenan-induced muscles from oxidative stress-induced damage and inactivate NF-ĸB-mediated proinflammatory cytokine expression.

As a pain-producing stimulus, carrageenan, an extract from seaweed, has been widely used in assessing inflammatory pain in various animal models. It induces gastrocnemius injury that is recognized as a simplified model of muscle inflammation characterized by delayed-onset muscle soreness, a reduction in grip force, and pain [30]. In the present study, we used intramuscular injections of carrageenan to produce muscular inflammation for mimicking DOMS in rodents.

Inflammation and oxidative stress have been implicated in the pathogenesis and development of various acute muscle damages [31,32]. Carrageenan has been shown to elicit a local inflammation when injected into muscles, as evidenced by the accumulation of leukocytes that began 2 h post-injection and lasted for the next 8 h [33]. In the present study, triptolide significantly decreased muscular pro-inflammatory cytokine production and inflammatory cell filtration 6 h after administration, as well as the lipid hydroperoxide level, which is a specific biomarker of oxidative stress in carrageenan-induced muscle damage. Free radical-initiated oxidative stress has been reported to participate in the pathogenesis of inflammatory disorders from various animal studies [34]. Among the free radicals, NO, a reactive nitrogen species, is one of the important free radicals participating in the pathogenesis and development of oxidative stress [35]. NO reacts with superoxide anion to form peroxynitrite, a highly cytotoxic free radical [36]. On the other hand, NO also plays a crucial role in regulating inflammatory response [37]. During inflammation, activating NF-ĸB enhances the transcription of pro-inflammatory mediators, such as NO, TNF-α, and IL-1β [38]. However, NO sustains the activation of NF-ĸB in chondrocytes and macrophages [6,39]. Taken together, we suggest that the activation of NF-ĸB could be down-regulated by triptolide-associated iNOS inhibition in carrageenan-induced muscle inflammation.

Triptolide can inhibit iNOS expression via a reduction in MPO activity. NO reduces the catalytic activity of iNOS by competing with O_2_ at the catalytic site of the enzyme during catalysis, which generates inactive NOS–nitrosyl complexes [40,41]. However, MPO can be secreted from the primary granules and colocalized with iNOS in activated leukocytes [42], which up-regulates the catalytic activity of iNOS by preventing feedback inhibition of NO [43,44,45]. We showed that triptolide significantly decreased muscular iNOS expression and MPO activity, suggesting that inhibiting MPO activity is crucial in the protective effect of triptolide from carrageenan-induced acute muscle damage.

LDH and CPK are the fragments of the myosin heavy chain, both of which cannot cross the sarcoplasmic membrane barrier [46,47]. Therefore, increased serum concentrations of the two molecules are indicators of damage to muscle membrane [48]. Although it has been reported to have hepatoxicity [49], in the present study, triptolide at the therapeutic doses significantly decreased LDH and CPK serum levels without affecting GOT and GPT. Both are the biomarkers of hepatic function. We suggest that triptolide might have a clinical application for the patients with acute muscle inflammation.

## 5. Conclusions

In conclusion, we have shown that the traditional Chinese herb extract triptolide can attenuate muscle inflammation by inhibiting MPO-associated oxidative stress in carrageenan-induced DOMS in rats without obvious adverse events.

## Figures and Tables

**Figure 1 ijerph-19-16685-f001:**
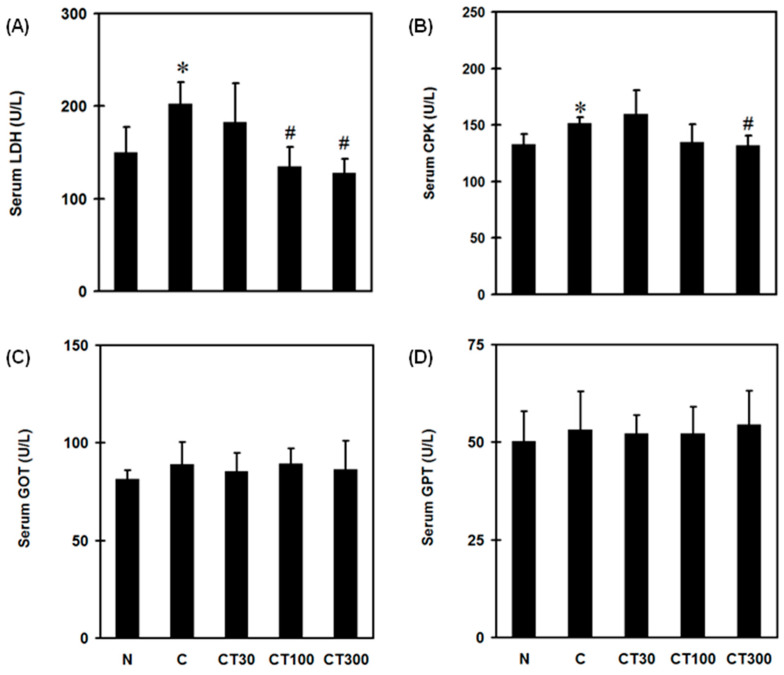
Effects of triptolide on serum biochemical changes. Rats were randomly divided into five groups. In the normal group (N), rats received 100 μL of saline by intramuscular (i.m.) and 500 μL of saline by intraperitoneal (i.p.) injections. In the carrageenan group (C), rats received 100 μL of 3% carrageenan (i.m.) and 500 μL of saline (i.p.). In the carrageenan and triptolide co-treatment groups (CT30, 100, 300), rats received 100 μL of 3% carrageenan (i.m.) and 500 μL of triptolide (30, 100, and 300 mg/kg, respectively) (i.p.). Serum LDH (**A**), CPK (**B**), GOT (**C**), and GPT (**D**) levels were measured 6 h after saline or carrageenan administration by a serum biochemical analyzer. Data are the means ± standard deviation (SD) (*n* = 6). * *p* < 0.05 compared with N group; ^#^
*p* < 0.05 compared with C group.

**Figure 2 ijerph-19-16685-f002:**
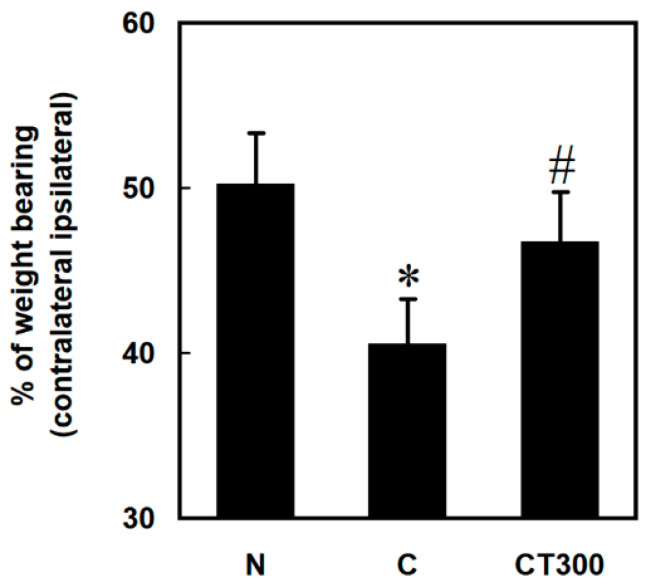
Effects of triptolide on pain levels. Rats were divided into three groups. In the N group, rats received 100 μL of saline (i.m.) and 500 μL of saline (i.p.). In the C group, rats received 100 μL of 3% carrageenan (i.m.) and 500 μL of saline (i.p.). In the CT300 group, rats received 100 μL of 3% carrageenan (i.m.) and 500 μL of triptolide (300 mg/kg) (i.p.). Animal pain assessment was conducted 6 h after saline or carrageenan administration. Percentage of weight bearing was normalized by the weight on the right limb divided by the total weights of the two hindlimbs Data are the means ± standard deviation (SD) (*n* = 6). * *p* < 0.05 compared with N group; ^#^
*p* < 0.05 compared with C group.

**Figure 3 ijerph-19-16685-f003:**
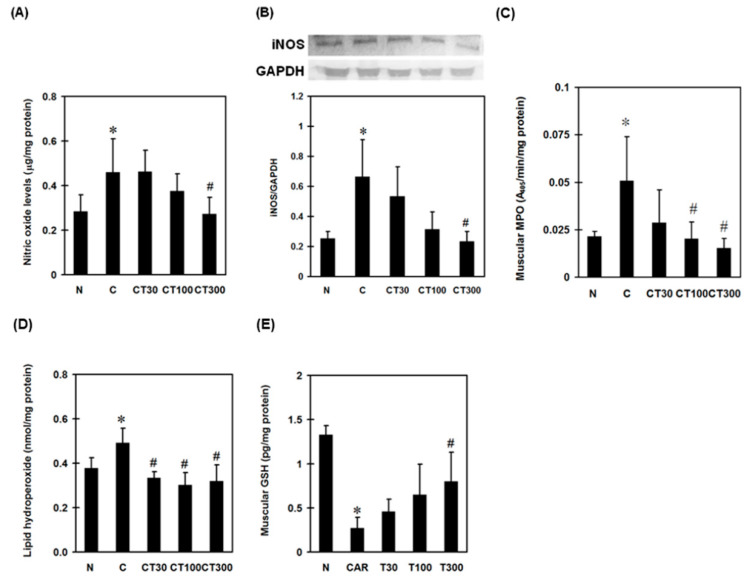
Effects of triptolide on muscular nitric oxide (NO) level, myeloperoxidase (MPO) activity, iNOS, lipid hydroperoxide, and glutathione (GSH) expression. Rats were divided into five groups. In group I (N group), rats received 100 μL of saline (i.m.) and 500 μL of saline (i.p.); group II (C group) rats received 100 μL of 3% carrageenan (i.m.) and 500 μL of saline (i.p.); and groups III-V received 100 μL of 3% carrageenan (i.m.) and 500 μL of triptolide (30, 100, and 300 mg/kg, respectively) (i.p.). Muscular NO production (**A**), iNOS (**B**), MPO (**C**), lipid hydroperoxide (**D**), and GSH (**E**) expression levels were determined 6 h after saline or carrageenan administration. Data are means ± standard deviation (SD) (*n* = 6). * *p* < 0.05 compared with N group; ^#^
*p* < 0.05 compared with C group.

**Figure 4 ijerph-19-16685-f004:**
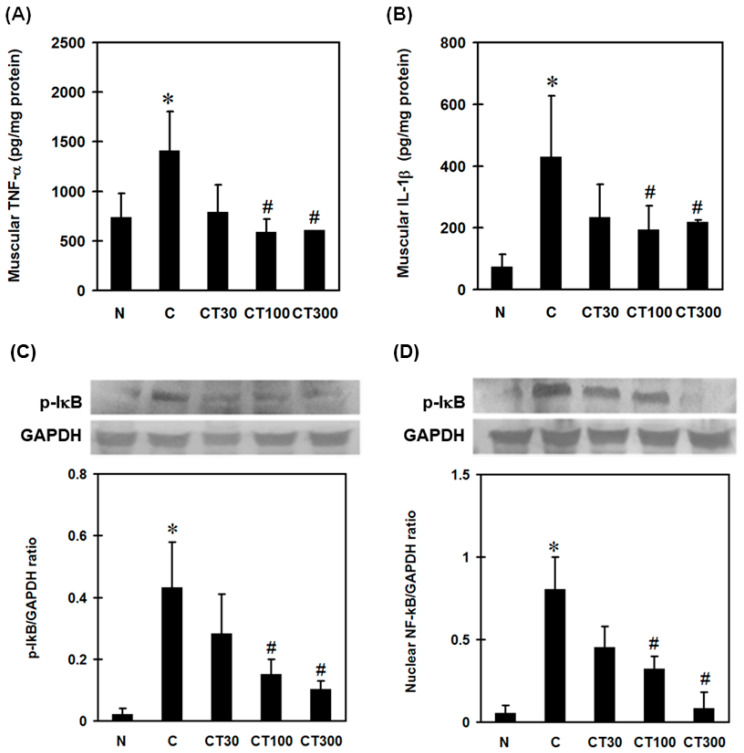
Effects of triptolide on muscular TNF-α, IL-1β, phospho-IĸB (p-IĸB), and nuclear NF-ĸB expression. Rats were divided into five groups of six. Group I (N group) rats received 100 μL of saline (i.m.) and 500 μL of saline (i.p.); group II (C group) rats received 100 μL of 3% carrageenan (i.m.) and 500 μL of saline (i.p.); and groups III-V received 100 μL of 3% carrageenan (i.m.) and 500 μL of triptolide (30, 100, and 300 mg/kg, respectively) (i.p.). Muscular TNF-α (**A**), IL-1β (**B**), p-IĸB (**C**), and nuclear NF-ĸB (**D**) were determined 6 h after saline or carrageenan administration. Data are means ± standard deviation (SD) (*n* = 6). * *p* < 0.05 compared with N group; ^#^
*p* < 0.05 compared with C group.

**Figure 5 ijerph-19-16685-f005:**
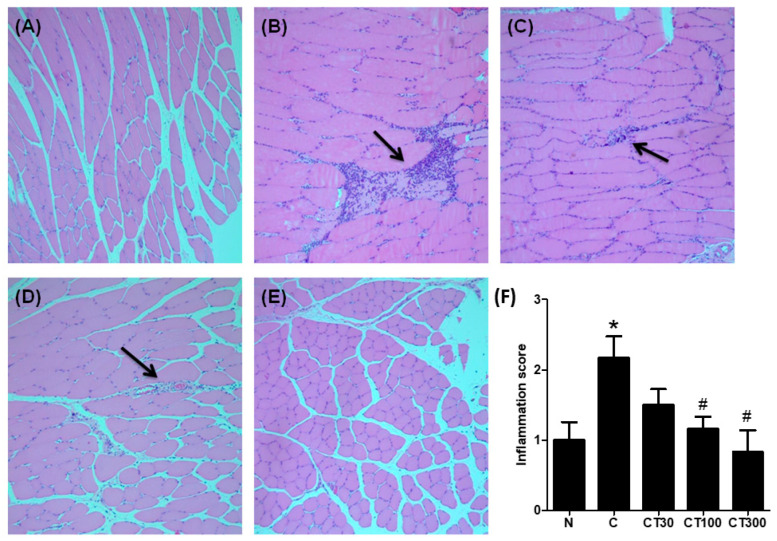
Effect of triptolide on histological changes in carrageenan-treated rats. Rats were divided into five groups of six. Group I (N group) rats received 100 μL of saline (i.m.) and 500 μL of saline (i.p.); group II (C group) rats received 100 μL of 3% carrageenan (i.m.) and 500 μL of saline (i.p.); and groups III-V received 100 μL of 3% carrageenan (i.m.) and 500 μL of triptolide (30, 100, and 300 mg/kg, respectively) (i.p.). Muscle tissues were collected from N group (**A**), C group (**B**), as well as 30 mg/kg (**C**), 100 mg/kg (**D**), and 300 mg/kg (**E**) of triptolide-treated groups. (**F**) Inflammation score. Data are means ± standard deviation (SD) (*n* = 6). * *p* < 0.05 compared with N group; ^#^
*p* < 0.05 compared with C group. Arrows indicate inflammatory cell infiltrates. (Magnification ×100).

## Data Availability

The data that support the findings of this study are available on request from the corresponding author.

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
