# Peer review of "Triptolide Attenuates Muscular Inflammation and Oxidative Stress in a Delayed-Onset Muscle Soreness Animal Model"

_ijerph, 2022, doi:10.3390/ijerph192416685_

Round 1

Reviewer 1 Report

The article “Triptolide attenuates muscular inflammation and oxidative stress in a delayed-onset muscle soreness animal model” by Che-Chia Hsu et al., evaluates the role of triptolide on carrageen induced acute muscle damage in rat model. By measuring the effect of triptolide on various inflammatory and oxidative stress markers, the authors have demonstrated that triptolide can attenuate carrageenan induced acute muscle inflammation. Overall, the article is well written. However, there are few suggestions which might further improve the quality of this research article.

1.     Carrageenan induced inflammation model does not represent exercise induced delayed onset muscle soreness (DOMS) model. There are several exercise induced animal models (e.g. PMID: 22431007; PMID: 35069187) which better represents DOMS. Authors should justify the reason for using carrageenan induced inflammation model.

In figure 3 and 4, western blot data should be replaced with a better-quality image.

Author Response

The article “Triptolide attenuates muscular inflammation and oxidative stress in a delayed-onset muscle soreness animal model” by Che-Chia Hsu et al., evaluates the role of triptolide on carrageen induced acute muscle damage in rat model. By measuring the effect of triptolide on various inflammatory and oxidative stress markers, the authors have demonstrated that triptolide can attenuate carrageenan induced acute muscle inflammation. Overall, the article is well written. However, there are few suggestions which might further improve the quality of this research article.

Response: Thank you very much for taking your precious time reviewing our manuscript and leaving the 2 critical points. We will respond to your valuable suggestions as following,

  1. Carrageenan induced inflammation model does not represent exercise induced delayed onset muscle soreness (DOMS) model. There are several exercise induced animal models (e.g. PMID: 22431007; PMID: 35069187) which better represents DOMS. Authors should justify the reason for using carrageenan induced inflammation model.

Response: Thank you very much for raising this point. We agree with your point. As stated in the Discussion section (Page 13, line 284-286), “Although carrageenan-induced inflammation model does not fully represent exercise induced DOMS model, carrageenan has been shown to elicit a local inflammation when injected into muscles as evidenced by accumulation of leukocytes that began 2 h post-injection and lasted for the next 8 h33. Due to the equipment constraints (treadmill or swimming pool), carrageenan-induced muscle soreness can be an alternative model to study inflammation and oxidative stress, which have been implicated in the pathogenesis and development of various acute muscle damages31,32” and “carrageenan induces gastrocnemius injury that is recognized as a simplified model of muscle inflammation characterized by delayed-onset muscle soreness, reduction in grip force and pain30.” (Page 13, Line 277-282). To test the anti-inflammatory effect of triptolide on muscular inflammation, we choose this model for mimicking DOMS in rats (Fig. 5).

  1. In figure 3 and 4, western blot data should be replaced with a better-quality image.

Response: Thank you for the suggestion. We have increased the resolution of each blot image and zoomed in figure 3 and 4. We suggest the 2 figures are suitable for presentation. Furthermore, the quantitated results of each band have been attached below each blot image (The bar graphs).

Reviewer 2 Report

In this manuscript, the authors reveal that triptolide can attenuate muscular damage by inhibiting muscular oxidative stress and inflammation in a carrageenan-induced rat DOMS model. The review is well written. I have following suggestions for authors to consider in revision.

Major concerns:

1.     please add the catalog number of all the reagents and antibodies.

2.     For fig.1, 3, 4. You should use two-way ANOVA rather than one-way ANOVA, as there are two factors Carrageenan and triptolide.

3.     For fig.1-4. you should add the statistical analysis for every group versus N group?

4.     For fig.4C and D. you should quantify p-IKB/NF-KB. What’s the phosphorylation site of p-IKB?

5.     For fig.5. You should quantify the inflammation index. You should stain and quantify the immune cells such as T cells (anti-CD3), neutrophil and macrophage (anti-CD11b) among five group.

Minor concerns:

1.     “Creatine-phospho-kinase” should be “Creatine phosphokinase”

Author Response

Comments and Suggestions for Authors

In this manuscript, the authors reveal that triptolide can attenuate muscular damage by inhibiting muscular oxidative stress and inflammation in a carrageenan-induced rat DOMS model. The review is well written. I have following suggestions for authors to consider in revision.

Response: Thank you very much for taking your precious time reviewing our manuscript and leaving the 6 critical points. We will respond to your valuable suggestions as following,

Major concerns:

  1. please add the catalog number of all the reagents and antibodies.

Response: Thank you for the suggestion. The catalog numbers of all the reagents and antibodies have been included in the section “Materials and Methods” with track changes.

  1. For fig.1, 3, 4. You should use two-way ANOVA rather than one-way ANOVA, as there are two factors Carrageenan and triptolide.

Response: Thank you for the suggestion. However, we think that two-way ANOVA is not necessary. In the statistical analysis, the among- and between-group comparisons were basically divided into 2 parts. The first part is the comparison between N and C, in which the factor is Carrageenan. The second part is the comparison among C, CT30, CT100, and CT300, in which the factor is triptolide. The 2 parts are independent comparisons. In each comparison, there was only 1 experimental factor, Carrageenan or triptolide. Therefore, we do not change our methodology. Hope you can understand our consideration.      

  1. For fig.1-4. you should add the statistical analysis for every group versus N group?

Response: Thank you very much for the suggestion. As our response to your comment No.2, the comparisons were divided into two parts. Based on the rationale, we thick that it is not necessary to compare the expression of every group with N group. It would be more important to evaluate the effect of triptolide on the DOMS model.

  1. For fig.4C and D. you should quantify p-IKB/NF-KB. What’s the phosphorylation site of p-IKB?

Response: Thank you for the suggestions. We have quantified p-IKB/NF-KB in fig.4C and D (The bar graphs below each blot images). The phosphorylation site of p-IKB is serine 32, which has been included in the section “Materials and Methods” (Page 3, line 118).  

  1. For fig.5. You should quantify the inflammation index. You should stain and quantify the immune cells such as T cells (anti-CD3), neutrophil and macrophage (anti-CD11b) among five group.

Response: Thank you for this valuable suggestion. The inflammation score has been attached in fig. 5F, and the related method information had been updated (Page 4, Line 150-152) 

Minor concerns:

  1. “Creatine-phospho-kinase” should be “Creatine phosphokinase”

Response: Thank you very for raising this issue . We have corrected Creatine-phospho-kinase to Creatine phosphokinase.
